# Pathogenicity Prediction of Gene Fusion in Structural Variations: A Knowledge Graph-Infused Explainable Artificial Intelligence (XAI) Framework

**DOI:** 10.3390/cancers16101915

**Published:** 2024-05-17

**Authors:** Katsuhiko Murakami, Shin-ichiro Tago, Sho Takishita, Hiroaki Morikawa, Rikuhiro Kojima, Kazuaki Yokoyama, Miho Ogawa, Hidehito Fukushima, Hiroyuki Takamori, Yasuhito Nannya, Seiya Imoto, Masaru Fuji

**Affiliations:** 1Computing Laboratories, Fujitsu Research, Fujitsu Ltd., Kawasaki 211-8588, Kanagawa, Japan; 2Division of Hematopoietic Disease Control, The Institute of Medical Science, The University of Tokyo, Tokyo 108-8639, Japan; 3The University of Tokyo Hospital, The University of Tokyo, Tokyo 113-8655, Japan; 4Division of Health Medical Intelligence, Human Genome Center, The Institute of Medical Science, The University of Tokyo, Tokyo 108-8639, Japan

**Keywords:** gene fusion, structural sariant, genomic medicine, explainable artificial intelligence, knowledge graph

## Abstract

**Simple Summary:**

Cancer genome analysis often reveals structural variants (SVs) involving fusion genes that are difficult to classify as drivers or passengers. Obtaining accurate AI predictions and explanations, which are crucial for a reliable diagnosis, is challenging. We developed an explainable AI (XAI) system that predicts the pathogenicity of SVs with gene fusions, providing reasons for its predictions. Our XAI achieved high accuracy, comparable to existing tools, and generated plausible explanations based on pathogenic mechanisms. This research represents a promising step towards AI-supported decision making in genomic medicine, enabling efficient and accurate diagnosis.

**Abstract:**

When analyzing cancer sample genomes in clinical practice, many structural variants (SVs), other than single nucleotide variants (SNVs), have been identified. To identify driver variants, the leading candidates must be narrowed down. When fusion genes are involved, selection is particularly difficult, and highly accurate predictions from AI is important. Furthermore, we also wanted to determine how the prediction can make more reliable diagnoses. Here, we developed an explainable AI (XAI) suitable for SVs with gene fusions, based on the XAI technology we previously developed for the prediction of SNV pathogenicity. To cope with gene fusion variants, we added new data to the previous knowledge graph for SVs and we improved the algorithm. Its prediction accuracy was as high as that of existing tools. Moreover, our XAI could explain the reasons for these predictions. We used some variant examples to demonstrate that the reasons are plausible in terms of pathogenic basic mechanisms. These results can be seen as a hopeful step toward the future of genomic medicine, where efficient and correct decisions can be made with the support of AI.

## 1. Introduction

In cancer genome medicine, interpreting the vast amount of variant data from the whole-genome sequencing of individual patients is a labor-intensive task and a bottleneck in the advancement of genomic medicine. In addition to single nucleotide variants (SNVs), structural variants (SVs) involving gene fusions have also gained significant attention. This is because fusion genes are observed in 20% of cancers [1], and their interpretation is crucial, as they play key roles in cancer development and progression.

Many SVs, including gene fusions, are detected by next-generation sequencers (NGS) and the downstream analysis of callers such as Manta [2]. However, not all gene fusions are pathogenic. To address this, several AI tools have been developed to predict the pathogenicity of the SVs containing fusion genes [3,4,5]. Nevertheless, high accuracy alone is insufficient for AI prediction as there is a risk of errors, and physicians must validate AI estimations before making a diagnosis. This validation process is time consuming and requires expert knowledge, which can be a significant burden on healthcare professionals. Thus, it is necessary to alleviate the burden of validating the estimation results.

The recently developed explainable AI (XAI) technology enables users to validate prediction results by providing explanations for AI estimations. Recently, XAI has been actively studied in various medical fields [6]. We have previously developed an XAI to predict the pathogenicity of SNVs [7].

Here, based on our previous XAI technology, an XAI was developed to predict the pathogenicity of SVs involving fusion genes, and to explain the prediction results (Figure 1). To achieve this, we expanded the knowledge graph (KG) (Figure 1, middle) by adding information related to the fusion genes, thus improving the algorithm and utilizing a large language model (LLM) [8]. The prediction accuracy of our XAI was as high as that of the existing tools. Moreover, the predictions provided reasonable explanations for pathogenic mechanisms in multiple cases. This enabled us to explain the reasons behind the predictions, making it more efficient for physicians to interpret novel prediction results. This provides important clues and simplifies the interpretation process.

## 2. Materials and Methods

### 2.1. Knowledge Graph

The research and development effort for XAI-applicable genomic medicine is ever-growing. While we can easily obtain desirable data using the LLM, maintaining the large amount of existing data remains difficult. We approach this problem by converting the data into a KG. Briefly, we constructed our KG using an ontology based on the Resource Description Framework (RDF) [9], developed by the open community med2rdf [10]. This community is led by Kyoto University and the Database Center for Life Sciences. The proposed XAI can learn to classify graphs that consist of nodes and edges. Then, it can learn the subgraph of the KG as is.

The size of our KG was 15,563,273,478 triples. This is larger than the 103 million triples in DBpedia [11], which is a well-known large-scale KG. This KG was constructed to estimate the pathogenicity of the SNV and short indels. We extended this KG to estimate the pathogenicity of fusion genes. The following extended data were used: Cosmic Fusion Export (v97; 86,506 records; 140,160 triples) [12], UCSC Gene Pfam (70,063 records; 935,892 triples) [13,14], UTR length from RefSeq (19,330 records; 154,640 triples) [15], TargetScan (235,109 records; 260,006 triples) [16], Mitelman (52,196 records; 382,476 triples) [17,18], Gene Ontology (317,823 records; 33,277 triples) [19,20], TumorFusions (24,569 records; 586,429 triples) [21], and ChimerDriver (6798 records; 33,865 triples) [3], for a total of 2,526,745 triples.

### 2.2. Explainable AI (XAI)

Our XAI, DeepTensor [22], learns to classify graphs that consist of nodes and edges. During learning, feature graphs with correct answers, which are positive or negative, are required. It outputs the predicted answer, and the added feature graph contributes to the weights of the edges when given a feature graph. Generally, black-box machine learning makes it difficult to explain the basis of the answers. Our XAI can achieve this despite being a black-box machine learning, as it was enhanced by an explainer based on a local interpretable model-agnostic explanation (LIME) [23]. LIME is an induced explaining model that explains the basis using a simple model that approximates the original learning model for ease of interpretation.

Most physicians find it difficult to understand the basis of a feature graph using the contribution weights of edges. The graph was first divided based on this point of view. Each subgraph contains human-comprehensible events and total contribution weights. Second, we converted the subgraph into natural language using the LLM. Converting a subgraph to natural language without a vocabulary of the RDF ontology appeared difficult, even for an LLM. Finally, the gpt-35-turbo-16k [24] was provided with some conversion cases (Figure 2) depending on the point of view, and it converted the subgraph well.

The inductive basis of statistics via machine learning requires deductive evidence for better acceptance by physicians. An evidence generator was developed that outputs a summary and key points regarding the mechanisms by which each keyword is related to the target fusion gene. The method was as follows: each prompt template was prepared in advance (Figure 3) to provide a summary and key points using the LLM from each point of view. When a fusion gene name, keyword, and keyword type were provided, the abstracts related to these keywords were retrieved from PubMed using Azure OpenAI’s text-embedding-ada-002 [25]. Gpt-4 generates a summary and key points from the template and abstracts.

### 2.3. Our Explainable Learning Model

Two learning models were prepared, one for explainability and the other for benchmarking. For benchmarking, data leakage is unfavorable for performance measurements. However, it is favorable for explainability to show evidence that is roughly equivalent to the answer as a basis, even if it causes data leakage. We chose learning features to allow data leakage as our explainable learning model and learning features to avoid data leakage as our benchmark model.

#### 2.3.1. Correct Answer Set: Dataset 1

TumorFusions [21] were used as the correct answer set. This dataset contains 20,731 fusion genes detected in cancer cells and 3838 fusion genes from normal cells obtained from patients with cancer registered in the TCGA [26]. Fusion genes with proven drug responses were selected from cancer cells as a positive answer because we could understand the mechanisms of pathogenicity. Additionally, we input the related paper to the LLM to determine whether the drug affected the consequences of the fusion genes because LLMs possess reviewing abilities for papers [27]. Some meanings of proven drug responses based on the evidence level in the guidelines [28] were defined as follows:The target text describes approved drugs in Japan for $DISEASENAME.The target text describes FDA-approved drugs for $DISEASENAME.The target text is referenced by guidelines about $DISEASENAME.The target text describes highly statistically reliable clinical trials/meta-analyses and consensus among experts on $DISEASENAME.The target text describes FDA-approved drugs for other cancer types.The target text describes highly statistically reliable clinical trials/meta-analyses and consensus among experts regarding other cancer types.The target text describes small-scale clinical trials that have shown usefulness regardless of cancer type.The target text describes the usefulness shown in case reports regardless of cancer type.The usefulness of target text has been reported in preclinical studies (in vitro and in vivo).

$DISEASENAME is replaced by the patient’s disease name.

Gpt-4-32k was filtered from 20,731 fusion genes to 269 fusion genes that satisfied at least one of the above conditions. These were selected as the positive answers. Generally, machine learning does not yield good results if the correct answer is biased to one side. Subsequently, 807 fusion genes were randomly selected as the negative answers, three times the number of positive ones from normal cells.

#### 2.3.2. Features

XAI generates feature graphs from our KG when given a feature definition consisting of a set of paths and a correct answer set. The features of the explainable learning model are as follows:The amount of literature regarding the same fusion genes in Mitelman;The number of entries about the same fusion genes in COSMIC;The sequence of domains registered with Pfam on each gene;The lengths of both UTRs;TargetScan-registered miRNAs that affect UTR;Families of miRNAs;Whether each domain is within the breakpoint;

Where the numeric values are converted into graphs representing the two most significant digits because overlearning occurs as it is. As the number of digits increases, the structure increases in this model. This is expected to determine the relationship between large and small.

The number of database entries may inhibit the learning of other features because it contains almost the same information as the answer. To address this, parts of these features were not inserted. This allows it to learn feature graphs without the number of database entries, and allows it to learn feature graphs with the number of database entries.

### 2.4. Benchmark

The model was benchmarked against the state-of-the-art models, and another model was created for comparison with Lovino’s model (ChimerDriver) [3]. The same correct answer set was used as the model for learning and testing, but with different features.

#### 2.4.1. Existing Methods

Several tools have been developed to predict the pathogenicity of gene fusions.

ChimerDriver [3] employs a neural network architecture that integrates information on transcription factors, gene ontologies, microRNAs, functional information (labels of either oncogenes, driver genes, tumor suppressors, or others), and structural information (retained percentage and strands).

DEEPrior [4] uses a deep learning algorithm, specifically a convolutional neural network (CNN), to process the amino acid sequences of gene fusions. The features utilized include the sequences themselves, and the properties derived from these sequences, such as gene functionality, description, and protein length.

Pegasus [29] employed gradient tree boosting (GTB), a machine learning method that combines the three following techniques: gradient descent, ensemble learning (boosting), and decision trees. This approach is known to achieve considerable accuracy, and is relatively easy to use. Pegasus uses information from protein domain annotations (preserved or lost).

OncoFuses [5] uses Naïve Bayes, which calculates a simple additive score from many features based on the statistics of the training data. It uses features such as the promoter, 3′UTR, the protein interaction interface (PII), and functional profiles.

As ChimerDriver outperformed the other four models [3], we focused our comparison exclusively on ChimerDriver.

#### 2.4.2. Learning Set: Dataset 2 and Dataset 3

Dataset 2 had 1059 gene fusions registered with COSMIC as positive (pathogenic), and 706 gene fusions selected by Babiceanu et al. [30] registered as negative (non-pathogenic). The latter fusion genes were collected from the normal cells of various organisms and used as negative controls.

Dataset 3 had 2623 fusion genes made from TCGA by Gao et al. as positive, and another set of 2254 fusion genes selected by Babiceanu et al. as negative. Both Dataset 2 and Dataset 3 were obtained from the site of ChimerDriver [3].

#### 2.4.3. Method of Comparing the Performance of the Models

To evaluate the ability of XAI to provide accurate answers for unknown data, we followed the standard practice of using separate datasets for training and testing. This approach ensures that the accuracy of XAI can be properly measured on previously unseen data.

Cross Validation

The first method used to measure the performance was a 10-fold cross-validation. In this method, we train a portion of the set of correct answers and measure the estimation performance for the remaining set of correct answers. Thus, the set of correct answers was divided into 10 parts, the estimation performance was measured for each partition, and the average of the 10 estimation performances was used as the performance of the method.

2.Holdout Validation

The second method for measuring the performance is holdout validation. Using this method, the two following datasets were prepared: a training set and a test set. The training set was used for model training, and the estimation performance was measured using the test set. This method helps assess how well the model is likely to predict data with different characteristics.

#### 2.4.4. Our Benchmark Learning Model

For a fair comparison, a model was created to avoid data leakage. The feature of our benchmark learning model is the feature graph, as in our explainable learning model; however, it excludes the number of database entries. The features of our benchmark learning model are as follows:Sequences of domains registered with Pfam for each gene;Lengths of both UTRs;TargetScan-registered miRNAs that affect UTR;Families of miRNAs;Whether each domain was within the breakpoint.

## 3. Results

This section discusses the benchmark results obtained using our model, and compares them with those of state-of-the-art tools. Next, we present cases in which XAI outputs an explanation of its predictions.

### 3.1. Evaluation of Prediction Accuracy

#### 3.1.1. Evaluation using Cross-Validation

To evaluate our model, we used the benchmark datasets used by other models. Our model was trained on these datasets, and two types of evaluations were conducted as follows:

As the first evaluation method, following ChimerDriver [3], a 10-fold cross-validation was performed using dataset 2. The results showed that our model achieved an accuracy of 0.98, which is the same as that of ChimerDriver (0.98).

#### 3.1.2. Evaluation Using Holdout Validation

For the second evaluation method, following ChimerDriver, Dataset 1 was used as the training dataset, and Dataset 3 was used as the test dataset. We evaluated the performance using the F1 score, which is a widely used index, defined as the harmonic mean of sensitivity and positive predictive value (PPV) or precision. In the test set, the F1 score of the proposed model was 0.845, slightly outperforming ChimerDriver’s F1 score of 0.832.

### 3.2. Evaluation of the Explanation of Prediction

Regarding the explanatory capabilities of the XAI, we present its output for some cases obtained from the COSMIC database [12]. For each case, the validity was evaluated from both the physicians’ and engineers’ perspectives (details are provided in the Discussion section).

For the evaluation, the model was trained using Dataset 1 (refer to the Materials and Methods section). Although other datasets were used for tool comparison, we determined that the model trained on Dataset 1, which was created to enrich the driver variant fusion genes, was appropriate as a clinical model. The key point here is to assess how well the model can provide explanations that are helpful for clinical decision making or further study.

#### 3.2.1. Case 1: KIF5B::RET Fusion

Example 1 is a case of KIF5B::RET (COSMIC Fusion Mutation ID: COSF1242, Figure 4). A sample was obtained from a patient with lung adenocarcinoma [31]. Among the features used, the Pfam protein domain, the number of references, and UTR length were output as the basis of explanation. The “Pathogenicity Score” (percentage) at the top of Figure 4 indicates the predicted pathogenicity of the case. For the two genes in the fusion gene, the complete protein structure of the 5′ gene is shown in the first row, and that of the 3′ gene is shown in the second row. Grey regions represent the entire protein, whereas other colors, such as green and red, indicate the domain regions. The numbers below represent the amino acid residue positions. The lollipops [32] represent the breakpoint positions, with red indicating the predicted reference cases and blue indicating the known breakpoint cases present in the DB. The size (radius) of the Laplace circles was proportional to the logarithm of the frequency. Below, the explanations output from XAI are shown in a bulleted list in descending numerical order. The numbers in parentheses are the explanation scores, which range from 0 to 1, with higher values indicating more important features that influence the determination. The threshold was set at 0.05.

The keywords in blue are hyperlinked, and when you click on them, you get a description of how the features that contribute to pathogenicity on the next screen are generated using the LLM. Part of the results are shown in Figure 5.

#### 3.2.2. Case 2: BCR::JAK2 T Fusion

Example 2 shows a case of BCR::JAK2 (COSMIC Fusion Mutation ID: COSF757, Figure 6). Samples were obtained from hematopoietic and lymphoid tissues. The BCR-JAK2 fusion gene was found in hematopoietic and lymphoid acute myeloid leukemia (AML), resulting from a t(9;22)(p24;q11) translocation [33]. As shown in Figure 6, seven references and two kinase domains are used as elements of the explanation. See the description in Example 1 for information regarding Figure 4.

Figure 6 shows an explanation of the featured domain Pkinase and the pathogenicity of the fusion gene BCR::JAK2.

XAI explained how the JAK2 tyrosine kinase domain is linked to pathogenicity, complementing the information on the STAT5 pathway. Figure 7 shows the results generated using the LLM as the explanation in sentences.

#### 3.2.3. Case 3: KIAA1549::BRAF Fusion

Example 3 is a case with the KIAA1549::BRAF fusion gene (COSMIC Fusion Mutation ID: COSF481, Figure 8) [34]. The tumor was located in the thalamus.

For Example 3, the selected case was from the dataset that included genes with miRNA features that were common in the overall data and where miRNAs were actually output in the explanation. In many cases, only one or two miRNAs were identified. See the description in Example 1 for information regarding Figure 8.

There were twenty-four references, two lengths of UTRs, kinase domains, and several miRNAs as elements of the explanation.

Figure 9 shows the results generated using the LLM as the explanation in sentences.

#### 3.2.4. Case 4: IKZF1::LRBA Fusion

Example 4 is a case with the IKZF1::LRBA fusion gene (Figure 10). The fusion was detected in the sample TCGA-D8-A27H-01A-11R-A16F-07 in TCGA (Breast invasive carcinoma (BRCA)). In Figure 10, two “zf-C2H2” domains, three references, miRNA, and the length of UTRs are included as elements of the explanation. Figure 11 shows the results generated using the LLM as the explanation in sentences.

#### 3.2.5. Case 5: APOE::ALB Fusion

Example 5 is a case with the APOE::ALB fusion gene (Figure 12) from non-oncogenic fusion data [30]. The sample is derived from normal cells in the liver, and is considered as a gene-neutral type fusion event. In Figure 12, serum albumin domains, apolipoprotein domain, and two lengths of UTRs are shown as elements of the explanation. Figure 13 shows the results generated using LLM as the explanation in sentences.

## 4. Discussion

Regarding the accuracy of the prediction, based on the results of both cross-validation and separate training/test datasets, our model performed slightly better than the state-of-the-art ChimerDriver tool. Similar to ChimerDriver, we employed miRNA information in our model.

As for the evaluation of explanatory capability, we showed three output examples.

In Example 1 (KIF5B::RET gene fusion, Figure 4), the gene fusion score was 100%, and the gene was predicted to be pathogenic. To explain the reason for this prediction, three features are shown (protein kinase (Pkinase) as a protein domain, fifteen references, and the length of the 3pUTR, Figure 4). Our XAI outputs the important domain “Pkinase” in *RET*. Because this is the most important feature of the pathogenicity of this fusion protein [31], the results are promising. One important feature missing in the explanation is the coiled-coil domain, with which the fusion of RET and KIF5B is maintained, along with the kinase domain [35]. This domain promotes homodimerization, and leads to the activation of the cancer-causing tyrosine kinase domain. After the prediction, we examined the dataset and revealed that the training dataset downloaded from the Pfam database did not include information regarding the coiled-coil domain. Therefore, when the training data include more detailed information, the re-trained XAI can possibly explain pathogenicity with tailored fine information, such as the coiled-coil domain.

Next, for the sentence explanation, the “Key point” mentioned the RET tyrosine kinase domain and suggested its importance in NSCLC; the statement was obtained from the searched literature [35,36]. As KIF5B is considered a common fusion partner of RET in lung adenocarcinoma [36], the prediction of our XAI model is appropriate. These results may be helpful for further studies.

In Example 2 (BCR:JAK2 gene fusion, Figure 6), the gene fusion score was 100%, and the gene was predicted to be pathogenic. The resulting BCR-JAK2 protein comprises the coiled-coil domain from BCR attached to the JH1-tyrosine-kinase domain from JAK2 [37]. The BCR-JAK2 fusion protein is supposed to be constantly active due to a mechanism similar to that of the well-known BCR-ABL fusion protein [38]. The coiled-coil domain of BCR causes the fusion protein to form oligomers (clusters), which leads to the continuous activation of the JAK2 kinase domain. Our XAI output the important “Pkinase” domains (both JH1 and JH2) in *JAK2*. However, XAI failed to include the “Bcr-Abl_Oligo” domain in BCR in the explanation.

Next, for the sentence explanation (Figure 7), our XAI explained how the JAK2 tyrosine kinase domain is linked to pathogenicity, complementing the information on the STAT5 pathway. This suggestion is based on the literature [39] found on PubMed by our XAI.

In Example 3 (KIKIAA1549::BRAF gene fusion, Figure 8), the gene fusion score was 100%, and the gene was predicted to be pathogenic. The KIAA1549:BRAF fusion gene is considered a driver in pilocytic astrocytoma [40], and constitutively activates the MAP kinase pathway [41]. Our XAI output the important “Pkinase” domains in *BRAF*. The UTR information as the target of miRNA and its length are highlighted. Most of the shown miRNAs (miR-148 [42], miR-30 [43], and miR-202 [44]), except for miR-6838, are tumor suppressors targeting KIAA2549, suggesting that this fusion is pathogenic.

Next, for the sentence explanation (Figure 9), our XAI explained how the BRAF tyrosine kinase domain is linked to pathogenicity, complementing the information on the MAP pathway. The explanation is based on the literature [45,46] found on PubMed by our XAI.

In Example 4 (IKZF1::LRBA gene fusion, Figure 10), the gene fusion score was 100%, and this fusion gene was correctly predicted to be pathogenic. In contrast to the above examples of gene-activating events via gene fusion, this case was selected as a gene-inactivating (loss of function) event. IKZF1, known as IKAROS, is a zinc finger transcription factor, characterized by its DNA-binding domain that contains multiple zinc finger motifs of the C2H2 type [47]. These motifs are crucial for the ability of the transcription factor to bind DNA and regulate gene expression. *IKZF1* plays a significant role in the development and differentiation of B cells. Alterations in *IKZF1*, such as deletions, is implicated in the pathogenesis of B-progenitor acute lymphoblastic leukemia (B-ALL) [47,48]. Our XAI output successfully identified the zinc finger domains of zf-C2H2 as bases of the prediction.

Next, for the sentence explanation (Figure 11), our XAI explained how the zinc finger domains of IKZF1 are linked to pathogenicity, explaining that gene fusion may disrupt the normal DNA-binding function, leading to alterations of B-cell development, based on the text in the searched literature [47].

In Example 5 (APOE::ALB gene fusion, Figure 12), the gene fusion score was 0%, and this fusion was correctly predicted to be benign (non-pathogenic). As the APOE::ALB fusion was obtained from the non-oncogenic dataset, this case should be regarded as a gene neutral-type fusion event. Our XAI outputs the serum albumin and apolipoprotein domains in *APOE* as bases of the prediction. It is generally difficult to identify a domain that does not contribute to carcinogenesis. However, it is reasonable to assume that the XAI system has learned the characteristics of domains that are not typically associated with cancer.

Next, for the sentence explanation (Figure 13), our XAI explained the potential pathogenicity of the APOE::ALB fusion gene. The XAI clearly stated that no direct evidence linking this fusion gene to pathogenicity was found in the literature. It also suggests that the APOE::ALB fusion gene might have some potential impact on disease development through various mechanisms, such as alterations in the lipid metabolism.

The length of the UTR can contribute to the pathogenicity prediction [22]. However, apart from the stability of the UTR, length is not widely recognized as an important feature in the pathogenicity of each fusion gene. However, it is difficult to consider the biological and medical implications of each case. The biological meaning of this is unknown, but this range of lengths increased the score in the first case (KIF5B::RET) and the third case (KIKIAA1549::BRAF), indicating a reason for this decision. This section briefly describes the coding methodology. When the length of the UTR is 117, the number 117 should not be regarded as a strict number, but should range from 96 to 127. If 117 is expressed as a binary number, it is 1110101. This is coded in binary form in our coding to a length ranging from 1100000 to 1111111 in binary numbers using the first two digits, or 96–127 in decimal.

The current study does not aim to provide specific therapy options or medication recommendations, as these aspects are beyond the scope of our work. Our study aims to support identifying potential druggable candidates, rather than confirm their actual applicability. Considerable validation is still required to determine whether these drug candidates are truly applicable. Therefore, we are not yet at the stage of presenting actual medications; this research is still in an exploratory phase. In Japan, the Center for Cancer Genomics and Advanced Therapeutics (C-CAT), a public organization, is responsible for suggesting applicable medications and clinical trials based on panel test results. Therefore, for established cases, such functionality already exists, and our study does not intend to duplicate the functionality of existing resources, but rather complement them by providing insights into potential druggable targets.

In addition to focusing on the features crucial for decision making, explanations have been provided using the LLM. This approach is believed to make the explanations more understandable. There are two types of explanations in conventional XAI evaluation methods as follows: global explanations, which show how features contribute to predictions across an entire dataset, and local explanations, which show how predictions are determined for individual cases. Here, the explanations are output from the perspective of indicating which features were focused on making judgments for individual predictions to determine whether further investigation by a physician is necessary. Our fundamental framework, DeepTensor [23], adopts LIME [7], a well-known method for local explanations. Conventional techniques often provide explanations oriented toward training data. However, as mentioned in our previous paper [7], this study focused on the development of the model for clinical use. To this end, a mechanism was devised that incorporates additional explanations related to the underlying mechanisms that can be easily understood by physicians.

## 5. Conclusions

We developed the first machine learning model, XAI, to predict the pathogenicity of SVs in cancers involving gene fusions, and provide explanations for these predictions. The accuracy was as high as that of the existing models. Moreover, the explanatory functionality of the prediction results makes our model more convenient than other models. We obtained valid explanations for the basic pathogenic mechanisms in multiple cases.

These findings can be seen as a promising step towards the future of genomic medicine, where efficient and accurate decision making or further surveys can be realized with the support of AI.

## Figures and Tables

**Figure 1 cancers-16-01915-f001:**
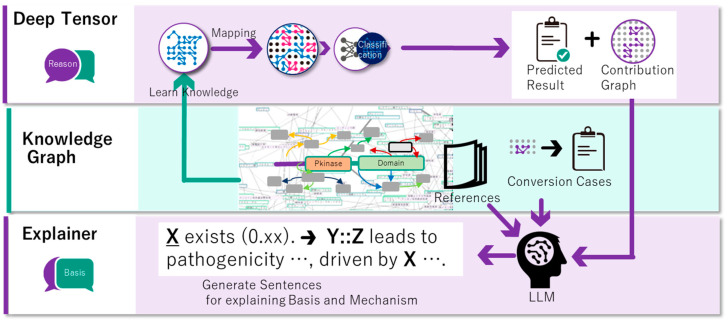
Overview of the explainable AI (XAI) methodology using the knowledge graph and deep tensor (fusion gene version). LLM outputs a simple sentence including a factor (X) and a summary including a mechanism of X.

**Figure 2 cancers-16-01915-f002:**
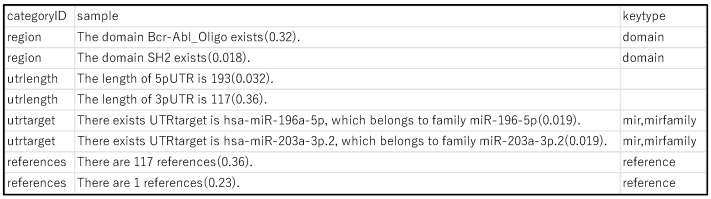
Conversion samples for each point of view.

**Figure 3 cancers-16-01915-f003:**
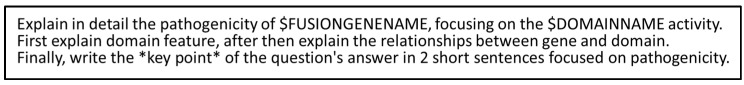
Sample template of prompt for gpt-4.

**Figure 4 cancers-16-01915-f004:**
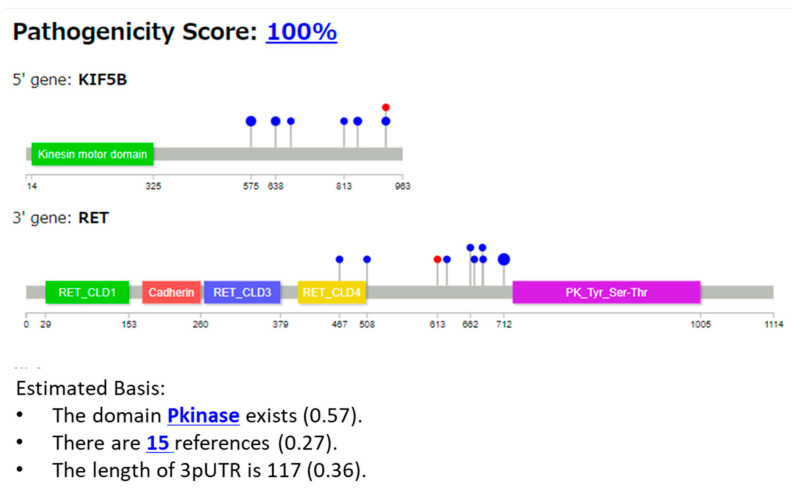
Protein structures of fusion genes of Example 1 (KIF5B::RET), and the explanation with features output from our XAI. Pkinase, protein kinase.

**Figure 5 cancers-16-01915-f005:**
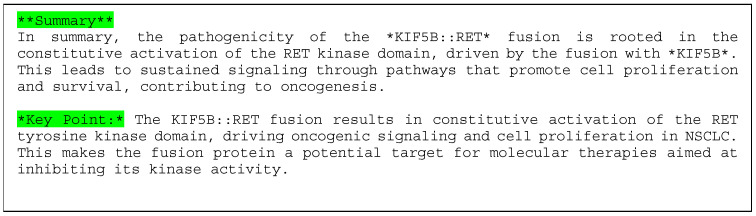
Explanation using sentences of Example 1 (KIF5B::RET).

**Figure 6 cancers-16-01915-f006:**
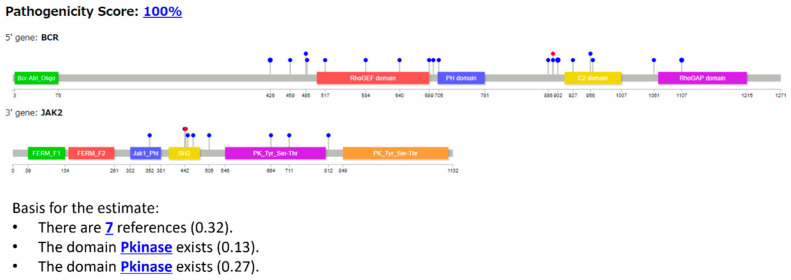
Protein structures of the fusion genes of Example 2 (BCR::JAK2), and the explanation using features output from our XAI. Pkinase, protein kinase.

**Figure 7 cancers-16-01915-f007:**
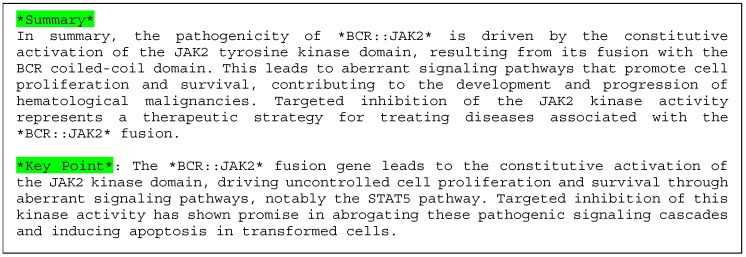
Explanation of the results obtained using the LLM for Example 2 (BCR::JAK2) by sentences.

**Figure 8 cancers-16-01915-f008:**
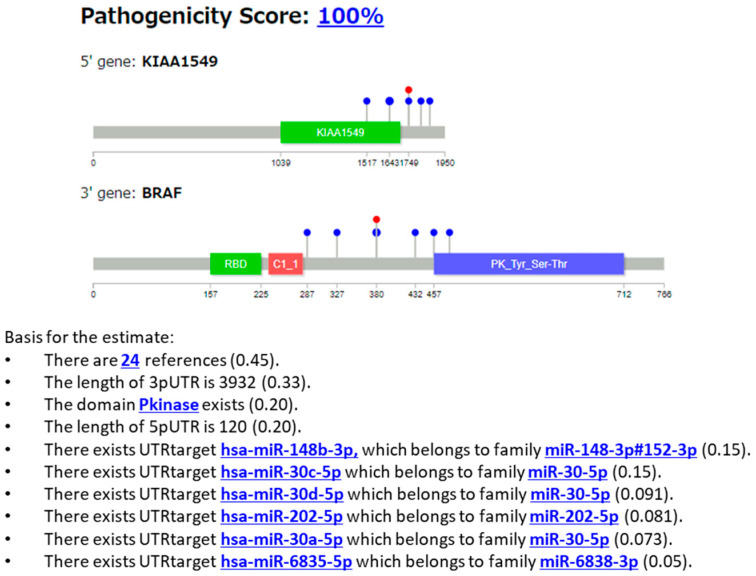
Protein structures of fusion genes of Example 3 (KIAA1549::BRAF), and the explanation using features output from our XAI. Pkinase, protein kinase.

**Figure 9 cancers-16-01915-f009:**
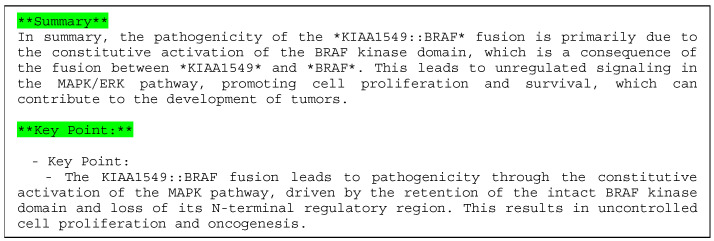
Explanation using sentences of Example 3 (KIAA1549::BRAF).

**Figure 10 cancers-16-01915-f010:**
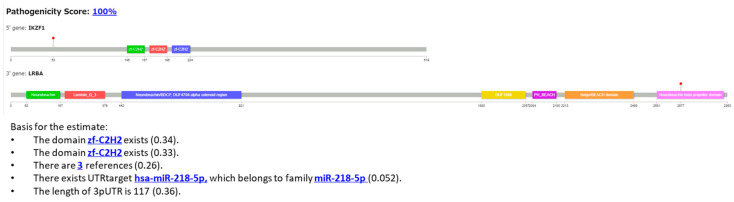
Protein structures of fusion genes shown in Example 4 (IKZF1::LRBA), and the explanation using features output from our XAI.

**Figure 11 cancers-16-01915-f011:**
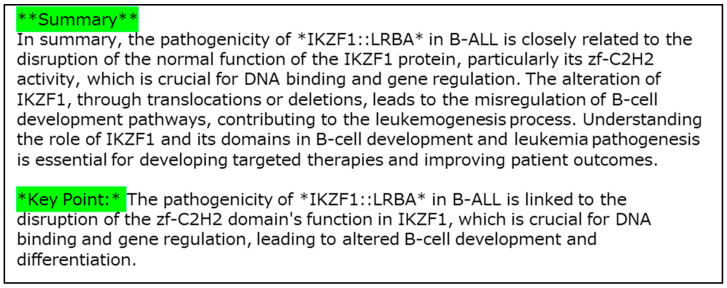
Explanation of the results obtained using LLM for Example 4 (IKZF1::LRBA) in sentences.

**Figure 12 cancers-16-01915-f012:**
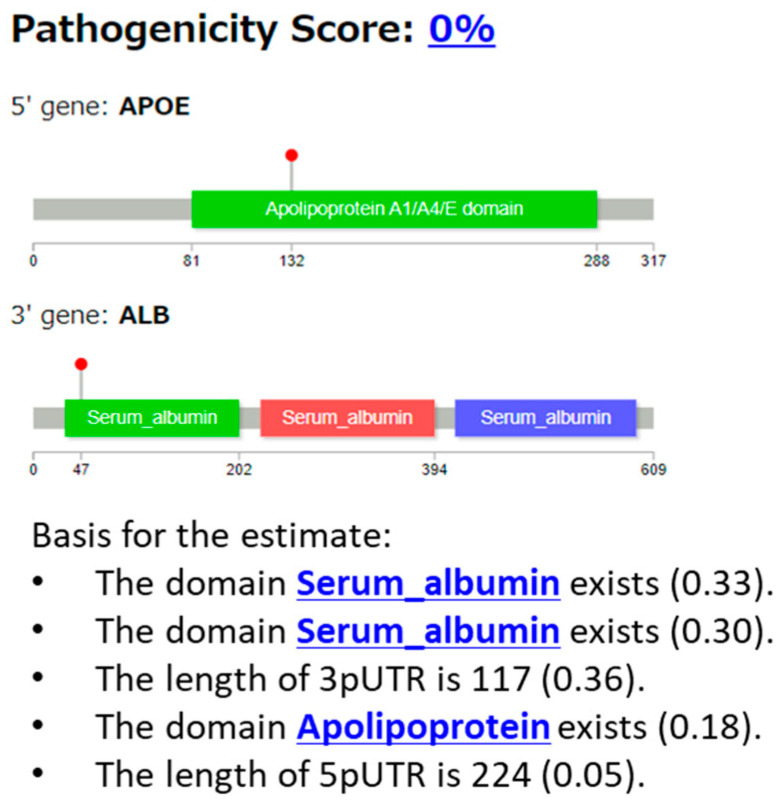
Protein structures of fusion genes of Example 5 (APOE::ALB), and the explanation using features output from our XAI.

**Figure 13 cancers-16-01915-f013:**
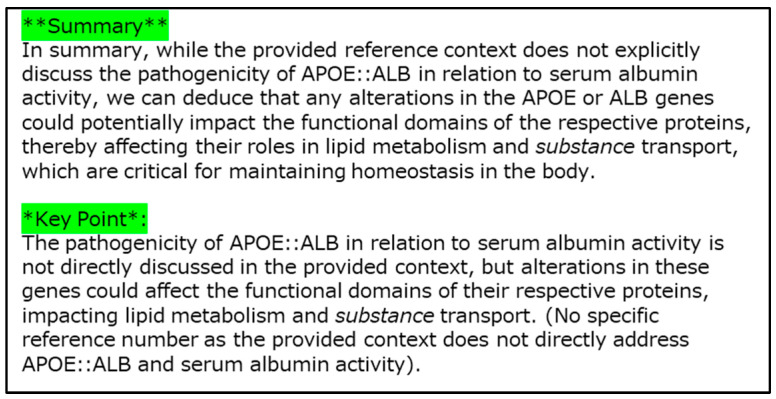
Explanation of the results obtained using LLM for Example 5 (APOE::ALB) in sentences.

## Data Availability

The data can be shared upon request. However, some data cannot be obtained because this technology needs to be commercialized.

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
