# Peer review of "Pathogenicity Prediction of Gene Fusion in Structural Variations: A Knowledge Graph-Infused Explainable Artificial Intelligence (XAI) Framework"

_cancers, 2024, doi:10.3390/cancers16101915_

Round 1
Reviewer 1 Report
Comments and Suggestions for Authors
This is a nice study, although it may require clarification of some nuances. Indeed, somatic mutations (including fusions) may be function-changing (potentially pathogenic) and function-neutral (passenger events). The latter may be gene-activating and gene-inactivating. Pathogenic fusions are not necessarily druggable for the time being, and matching them to existing drugs is actually a separate effort requiring additional thoughtful discussion. I would request for more clarity in discriminating between gene-activating and gene-inactivating events, as it is important to draw the line between oncogenes and tumor suppressor genes. Furthermore, only examples of activation of kinases are presented (which are well known and do not require artificial intelligence). I suggest to present more examples for other categories of fusions, e.g. function-inactivating and function-neutral events.
Comments on the Quality of English LanguageMinor editing required
Reviewer 2 Report
Comments and Suggestions for Authors
The application of AI methods has a great potential to improve cancer therapy. This manuscript describes the development of XAI for cancer and provided examples of the abilities of XAI in terms of notorious fusion genes. Although the manuscript is interesting and important, it needs some improvements before acceptance:
Lines 271, 285 and 304: The figure number is missing after ´´Figure´´. Please add.
Although the authors mention that clicking on the blue will provide the proper explanation, a brief definition of the ´´Pathogenicity Score´´ should be provided in the main text of the manuscript if possible (in particular, because all three examples show 100%).
Please specify the meaning of ´´Pkinase´´ as ´´protein kinase´´ in the text or in the respective figure captions.
Is the XAI also able to provide optimized therapy options or to suggest new drugs for the cancers with the described fusion genes which might be studied in clinical trials then? Please discuss.
Round 2
Reviewer 1 Report
Comments and Suggestions for Authors
The authors have revised the manuscript in a proper way.
Comments on the Quality of English LanguageEnglish is good.